# Developments in Post-Resuscitation Care for Out-of-Hospital Cardiac Arrests in Adults—A Narrative Review

**DOI:** 10.3390/jcm12083009

**Published:** 2023-04-20

**Authors:** Stephan Katzenschlager, Erik Popp, Jan Wnent, Markus A. Weigand, Jan-Thorsten Gräsner

**Affiliations:** 1Department of Anesthesiology, Heidelberg University Hospital, 69120 Heidelberg, Germany; erik.popp@med.uni-heidelberg.de (E.P.); markus.weigand@med.uni-heidelberg.de (M.A.W.); 2Institute for Emergency Medicine, University Hospital Schleswig-Holstein, 24105 Kiel, Germany; jan.wnent@uksh.de (J.W.); jan-thorsten.graesner@uksh.de (J.-T.G.); 3Department of Anesthesiology and Intensive Care Medicine, University Hospital Schleswig-Holstein, Campus Kiel, 24105 Kiel, Germany; 4School of Medicine, University of Namibia, Windhoek 9000, Namibia

**Keywords:** post-resuscitation care, out-of-hospital cardiac arrest, resuscitation, intensive care

## Abstract

This review focuses on current developments in post-resuscitation care for adults with an out-of-hospital cardiac arrest (OHCA). As the incidence of OHCA is high and with a low percentage of survival, it remains a challenge to treat those who survive the initial phase and regain spontaneous circulation. Early titration of oxygen in the out-of-hospital phase is not associated with increased survival and should be avoided. Once the patient is admitted, the oxygen fraction can be reduced. To maintain an adequate blood pressure and urine output, noradrenaline is the preferred agent over adrenaline. A higher blood pressure target is not associated with higher rates of good neurological survival. Early neuro-prognostication remains a challenge, and prognostication bundles should be used. Established bundles could be extended by novel biomarkers and methods in the upcoming years. Whole blood transcriptome analysis has shown to reliably predict neurological survival in two feasibility studies. This needs further investigation in larger cohorts.

## 1. Introduction

Out-of-hospital cardiac arrest (OHCA) remains a high burden for society and caregivers, with an annual incidence between 67 and 170 per 100,000 inhabitants in Europe, whereas the percentage of survival is reported to be 8% [1,2]. Studies focusing on specific treatments and populations achieve higher survival rates, which might not reflect the true rates in these countries. Rates of good neurological outcome vary between 3% and 8% for national registries [3,4].

Post-resuscitation care for OHCA has been continually improved over the last few years, with the aim of improving survival with favorable neurological outcomes, according to the cerebral performance category (CPC) or modified ranking scale (mRS). The return of spontaneous circulation (ROSC) is dependent on known factors, such as: shockable rhythm [5,6], preexisting conditions [7], bystander cardiopulmonary resuscitation (CPR) [8,9] and early defibrillation [10]. Early recognition and call for help in patients with typical chest pain is still the main link in the chain of survival [11]. This is easier said than done, as it takes education to recognize a critically ill person and a system to receive the emergency call, send adequate help, give first aid advice and perform telephone-CPR if necessary [12]. If this link is missing, the chances of survival diminish.

Early restoration of circulation, whether spontaneous or assisted via extracorporeal cardiopulmonary resuscitation (eCPR), is one cornerstone to initiate post-resuscitation care in OHCA [13]. Prediction scores for ROSC [14] can help determine which patients might be suitable for eCPR strategies if conventional measures fail at a certain timepoint. A recent evaluation of a prediction score for good neurological outcomes after OHCA identified variables to be of relevance only until admission of the patient [15], indicating that initial circumstances and prehospital therapy are the main drivers of good neurological outcomes.

This review highlights important studies assessing the treatment strategies for post-resuscitation care in patients with OHCA.

### Narrative Review

We selected articles cited by recent guidelines on post-resuscitation care [16] and used backwards and forwards reference searching to identify relevant articles. No systematic review was conducted. Articles were considered eligible if they were published within the last couple of years, without applying a strict cut off. No restrictions regarding language, country or type of study (e.g., observational vs. randomized) were made. Studies assessing only in-hospital cardiac arrest were not considered to be eligible for this review.

## 2. Post-Resuscitation Care—Changes from 2015 to 2021

In 2021, the current joint guideline from the European Resuscitation Council (ERC) and European Society of Intensive Care Medicine (ESICM) for post-resuscitation care in an intensive care unit (ICU) was released [16]. Several changes and novelties were introduced; these are herein summarized and put into context with the current literature.

For the first time, ERC and ESICM are emphasizing on the admission of patients with non-traumatic OHCA to a dedicated cardiac arrest center (CAC) [17]. This is expected to ultimately reduce morbidity and mortality. Due to the logistics of CAC, early percutaneous coronary intervention (PCI), temperature management and computer tomography should be achieved.

In 2015, the guideline recommended a mean arterial pressure (MAP) to achieve a urine output of 1 mL/kg/h and normal or decreasing lactate levels. This changed to the avoidance of hypotension (MAP < 65 mm Hg), urine output of >0.5 mL/kg/h and normal or decreasing lactate levels in the 2021 guidelines. Further, individualized MAP targets should be considered to enable adequate organ perfusion.

Temperature control between 32 °C and 36 °C was specified to be constant for at least 24 h. In addition, avoidance of fever for at least 72 h was explicitly recommended. A routine use of prophylactic antibiotics was not recommended. As for prognostication, neuron-specific enolase (NSE) was further specified with a predefined cut-off value of >60 µg/L at 48 h and 72 h, indicating a poor neurological outcome.

### 2.1. Oxygenation/Ventilation

After ROSC, it is recommended that patients maintain normoxaemia with an SpO_2_ of 94–98% or PaO_2_ of 75 to 100 mm Hg [16]. Early titration of oxygen saturation was investigated in patients with prehospital ROSC transported by paramedics [18]. Herein, 425 patients were randomized to the intervention group, which targeted a SpO_2_ of 90% to 94%. Patients in the control group were targeted to a SpO_2_ of 98% to 100%. The primary outcome was survival to hospital discharge, and there were nine secondary outcomes such as hypoxia or survival with a CPC score of 1–2. Out of 647 eligible patients, 425 were included in the final analysis. The main proportion of patients were male with a median age of 66 and 64 years in the intervention and control groups, respectively.

Both groups had high rates of bystander-witnessed arrests, bystander CPR and shockable rhythm. Survival to hospital discharge was lower in the intervention group, touching the level of significance (odds ratio (OR) 0.68 [95% confidence interval [CI] 0.46 to 1.00]; *p* = 0.05). Secondary analysis showed a significantly higher odds ratio (OR) of hypoxic (SpO_2_ < 90%) episodes prior to ICU admission in the interventions group (OR 2.37 [95%CI 1.49 to 3.79]); however, the rates of rearrests and good neurological outcomes were similar. This study indicates that an SpO_2_ between 90% and 94% in patients with OHCA and ROSC can decrease survival rates.

In a different study, investigating how different oxygenation strategies in adults with OHCA and ROSC admitted to the ICU can have an impact, Schmidt et al. conducted a randomized controlled trial (RCT) [19]. These patients were randomized to receive either a restrictive (68 to 75 mm Hg) or liberal (98 to 105 mm Hg) oxygenation strategy for up to 5 days. A total of 789 patients were included in the final analysis. Baseline characteristics did not differ between the two groups. A higher proportion of men were included, with a mean age of 62 years, and with high rates of shockable rhythm (85%) and bystander CPR (89%). The primary outcome was death at 90 days or hospital discharge with a CPC score of 3–5, where no differences between both groups were found. Survival rates were 70% across both groups with a median CPC score of 1, indicating an a priori selected group of patients with a good probability of favorable neurological survival. Hypoxic events were not reported.

Early titration with lower targeted SpO_2_ values may decrease the chances of survival with good neurological outcomes, whereas in an ICU setting, a restrictive oxygenation strategy can be chosen without compromising the chances of survival. Mortality in correlation to PaO_2_ levels in critical ill patients have a U-shaped distribution [20]. Robba et al. performed a secondary analysis of the TTM_2_ trial, investigating the effects of hypo- and hyperoxemia within the first 72 h of admission on mortality and neurological status at 6 months. This multivariate Cox regression also showed a U-shaped distribution for PaO_2_ with an increased risk for mortality below 69 mm Hg (hypoxemia) and above 195 mm Hg (hyperoxemia); however, it found no correlation for poor neurological outcomes [21]. PaO_2_ was significantly lower in non-survivors for up to 32 h after ICU admission. In a different trial, the presence of hyperoxemia (PaO_2_ > 300 mm Hg) for a 1 h period increased the risk of poor neurological outcomes by 3% (relative risk 1.03 [95% CI 1.02 to 1.05]) [22].

Clinical trials investigating different oxygen levels in OHCA were conducted within the “safe” range of the curve, thus questioning if this U-shaped curve is still of relevance [23]. Several observational studies investigated the association between severe hyperoxemia (PaO_2_ > 300 mm Hg) and mortality or neurological outcomes. La Via et al. performed a meta-analysis across 13 observational studies and found that severe hyperoxemia is associated with a poor neurological outcome (OR 1.37 [95% CI 1.01 to 1.86]) and higher mortality (OR 1.32 [95% CI 1.11 to 1.57]) [24]. Furthermore, hyperoxemia during the first 36 h of ICU admission showed higher OR for poor neurological outcomes and mortality with 1.52 (95% CI 1.12 to 2.08) and 1.40 (95% CI 1.18 to 1.66), respectively.

Carbon dioxide is recommended to be kept from 35 to 45 mm Hg in patients requiring mechanical ventilation after ROSC. In a retrospective cohort study in Japan, normocapnia (35 to 45 mm Hg) and mild hypercapnia (45 to 55 mm Hg) during the first 24 h after ROSC were associated with a better neurological outcome [25]. This analysis was adjusted for multiple covariates which are known to influence outcomes in patients with OHCA. This study is in line with the guideline recommendations and further emphasizes the avoidance of hypocapnia in patients with ROSC. However, mechanical ventilation during CPR has shown to be unreliable as the overall deviation of the predetermined tidal volume was −21% [26]. This can result in higher PaCO_2_ levels after ROSC and should be closely monitored. Further, tidal volumes during chest compressions should be assessed if mechanical ventilation is used. An increase in the respiratory rate during OHCA does not seem to have an effect on PCO_2_, pH and ROSC rates [27]. Patients suffering from OHCA which achieved sustained ROSC and ICU admission receiving TTM were assessed for impacts of ventilator settings in the initial 72 h on mortality and neurological outcomes [28]. Analyzing ventilator parameters in 1848 patients, a higher respiratory rate was identified as being associated with a higher mortality rate and worse neurological outcomes.

### 2.2. Circulation/Vasopressors

As per the guideline recommendation, hypotension (MAP < 65 mm Hg) should be avoided. However, there is no clear definition on a MAP target above 65 mm Hg. To test the hypothesis of whether a higher MAP impacts the rate of all-cause mortality or CPC score 3–4, Kjaergaard et al. conducted a randomized controlled trial comparing two different MAP targets, 63 vs. 77 mm Hg in comatose adults after OHCA [29]. The baseline characteristics are described above in the study by Schmidt et al. To maintain the study’s specific MAP target, volume resuscitation, noradrenaline and dobutamine were used. No difference in mortality or unfavorable neurological outcomes between these two groups (32% vs. 34%, *p* = 0.56) was found. Secondary endpoints also showed no differences between the two groups, indicating that a higher MAP target does not improve overall survival and good neurological outcomes until 90 days.

To achieve and maintain this target, different medications can be used. In a multicenter observational study, 766 patients were included with post-ROSC shock. Of those, 481 received noradrenaline and 285 adrenaline to maintain an adequate MAP. Both groups had high rates of witnessed arrests (90%) and bystander CPR before EMS arrival (71%), with half of the patients having initial shockable rhythm.

Propensity score analysis and adjusted logistic regression found higher OR for all-cause mortality with 2.1 (95% CI 1.1 to 4) and 2.6 (95% CI 1.4 to 4.7), respectively. Unfavorable neurological outcomes (CPC 3-5) at discharge for patients receiving adrenaline had ORs of 4.3 (95% CI 2.2 to 8.3) and 5.5 (95% CI 3 to 10.3), respectively [30]. Although observational trials have certain biases, this finding impacts the further treatment of patients with hypotension after ROSC and supports the guideline recommendations on the usage of noradrenaline.

### 2.3. Glucose

As hyperglycemia is often observed after OHCA [31], the American Diabetes Association recommended a range from 140 to 180 mg/dL for critically ill patients, respectively, such as those admitted to an ICU after OHCA [32]. In order manage hyperglycemia, a continuous insulin infusion is favored over repetitive applications [16]. As hypoglycemia (<70 mg/dL) is harmful, blood glucose levels should be monitored closely [33].

### 2.4. Percutaneous Coronary Intervention

After stabilization of the patient, PCI should be performed in all patients with presumed or obvious cardiac causes of cardiac arrest [16]. This has shown to be beneficial in all patients with ECG changes in line with occlusion myocardial infarction (OMI). In patients with ROSC after OHCA without ST-segment elevation, Desch et al. investigated if an immediate PCI strategy would provide a benefit over a delayed PCI strategy [34]. This study investigated death from any cause at 30 days as the primary outcome and survival with severe neurological deficit as a secondary endpoint. In total, 554 patients were randomized to either an immediate or delayed angiography. With similar baseline characteristics, angiography was performed 2.9 h or 46.9 h after OHCA, respectively. This study found no benefit for an immediate angiography, as the primary endpoint was reached in 54% of the immediate group and 46% of the delayed group (hazard ratio [HR] 1.28 [95% CI 1.00 to 1.63]). This finding corroborates other trials that have failed to show survival benefits for an early coronary angiography strategy in patients after OHCA without ST-segment elevation [35,36,37]. In contrast to these RCTs, observational studies showed results in favor of early angiography [38].

Therefore, the current guidelines recommend early angiography to patients with hemodynamic instability or those of high suspicion for a cardiac cause [39,40].

### 2.5. Temperature Management/Fever Prevention

Temperature control has been discussed over the last two decades. The initial trials have shown a benefit when patients are initially cooled between 32 °C and 34 °C [41]; however, recent trials have found no difference in survival [42]. This is further corroborated by two meta-analyses from Aneman et al. and Sanfilippo et al. [43,44]. Aneman et al. conducted a Bayesian meta-analysis across seven RCTs, which identified no significant risk ratio (RR) for mortality (RR 0.96 [95%CI 0.82 to 1.04]) and unfavorable neurological outcomes (RR 0.93 [95%CI 0.84 to 1.02]) [43]. Sanfilippo et al. included eight RCTs which compared TTM to either actively controlled or uncontrolled hypothermia [44]. In the overall study group and actively controlled subgroup, no significant differences for mortality were found, with RRs of 1.06 (95% CI 0.94 to 1.20) and 0.97 (95% CI 0.90 to 1.04), respectively. In the passively controlled normothermia subgroup, a significant difference for mortality was observed (RR 1.31 [95% CI 1.07 to 1.59]). Temperature control to 32–34 °C did not influence neurological outcomes, with an overall RR of 1.17 (95% CI 0.97 to 1.41).

While there is a constant debate about patient selection and the time until hypothermia was initiated in those studies, current guidelines recommend the temperature control to be between 32 °C and 36 °C for at least 24 h, and further prevention of fever for 72 h. To understand if a longer TTM strategy after the initial 24 h is beneficial, Hassager et al. conducted a randomized controlled trial. In this study, patients after OHCA who remained comatose were randomized to receive TTM either 12 or 48 h after the initial 24 h. No difference was found in the composite outcome of death or CPC score 3–4 at 90 days (HR 0.99 [95% CI 0.77 to 1.26], *p* = 0.70) [45].

### 2.6. Antibiotics/Further Medications

Many patients aspirate during cardiac arrest and undergo mechanical ventilation during the initial course of ICU therapy. Due to temperature management and prevention of fever, clinical signs of infection can be missed. Current guidelines do not support the routine administration of antibiotics in patients with ROSC. If clinical signs of an infection persist after OHCA, these patients benefit from an antibiotic treatment. Patients requiring mechanical ventilation after OHCA are at risk of ventilator-associated pneumonia (VAP). Therefore, a preventive strategy with short-term antibiotic therapy was assessed by Francois et al. [46]. Patients with initial shockable rhythm receiving temperature management between 32 °C and 34 °C were randomized to either amoxicillin-clavulanate or placebo. This trial included 194 patients in both groups and resulted in a lower incidence of early VAP in the intervention group (19% vs. 34%). These results were mirrored in the significant reduction in VAP at any timepoint (23% vs. 39%). However, due to the small sample size, this did not impact mortality rates at day 28 (41% vs. 37%) or ICU length of stay. Even if this trial included only patients with shockable rhythm, these findings can be generalized to patients with non-shockable rhythm, as post-resuscitation care is similar between those groups.

A meta-analysis on prophylactic antibiotic use after OHCA did not show a significant increase in overall survival (OR 1.16 [95% CI 0.97 to 1.40]) or survival with good neurological outcomes (OR 2.25 [95% CI 0.93 to 5.45]) [47].

The usage of stress ulcer prophylaxis is recommended, even if it does not reduce mortality in ICU patients [48]. As patients after OHCA often are treated with antiplatelet and anticoagulant medication [49], they are at a higher risk of gastrointestinal (GI) bleeding. The usage of ulcer prophylaxis has shown to reduce the rates of GI bleeding in high-risk ICU patients [50].

### 2.7. Prognostication

Prognostication can already be started during advanced life support (ALS) using near-infrared spectroscopy (NIRS) to assess cerebral oximetry (rSO_2_). During CPR, increasing rSO_2_ values might be indicative of ROSC [51]. In an analysis of 52 patients with OHCA, initial rSO_2_ values were undetectable (<15%) and increased rapidly with near-normalization in patients who achieved ROSC. Patients with ROSC had a higher maximum value at 47% vs. 31% in those without ROSC [52]. Patients undergoing eCPR with a pre-cannulation rSO_2_ between 41% and 60% had the highest chance of good neurological survival (CPC 1 to 2) with almost 40% [53], while patients with a pre-cannulation rSO_2_ between 17% and 40% and above 60% had just over 20% good neurological survival. Besides rSO_2_, a rise in etCO_2_ was also found to be of prognostic value. Keeping in mind that etCO_2_ is prone to confounders such as respiratory rate, tidal volume and the type of airway device, the major limitation is that that an advanced airway must be applied.

After initial ICU treatment, the prognostication of neurological outcomes is one cornerstone in the early post-resuscitation care phase. This remains a challenge in post-resuscitation care as the false positive rate for the prediction of poor neurological outcomes should be 0%. Consequently, the reliable identification of patients who will not benefit from early invasive treatment strategies can be the main consequence of those tests. An inappropriate withdraw of life support must be avoided; this occurs due to a self-fulfilling prophecy bias [54,55]. Therefore, current guidelines recommend the investigation of prognostic biomarkers to be blinded. Withholding results from blood samples can be cumbersome and unethical in daily practice. To minimize these biases, an assessment of neurological outcomes is recommended to be performed at three-to-six months after OHCA [16].

Regarding biomarkers, NSE in combination with other tests is recommended to be the biomarker for neurological prognostication. A sole biomarker has substantial limitations for predictions, failing to safely identify patients with unfavorable neurological outcomes.

NSE is reported to decrease after 24 h in patients with good neurological outcomes and increase in those with unfavorable neurological outcomes between 48 and 96 h. This highlights that a single NSE value should be interpreted with caution; rather, its trend over the first few days is important. Further, hemolysis must be measured in order to detect false high NSE values [56]. Hemolysis was observed in more than half of the patients in up to 4 h after eCPR initiation; this rate decreased to approximately 10% after 24 h. In a retrospective cohort study of patients with OHCA and prehospital eCPR undergoing temperature management, there was a significant difference in NSE at 4 h between those with favorable neurological outcomes and those who died [57]. At 48 h, NSE could reliably distinguish between those with favorable and unfavorable neurological survival; this is while extracorporeal membrane oxygenation was still ongoing. The early withdraw of life support solely based on biomarkers is not recommended and should be avoided.

As a multi-modal approach, electroencephalogram (EEG) and cranial computed tomography (cCT)—besides others—are recommended. EEG has been thoroughly studied to predict brain function and prognosis after cardiac arrest [58]. It is described that EEG is suppressed immediately after OHCA and returns to normal voltage within 24 h in patients with good neurological outcomes [59]. CCT is used to evaluate the ratio between grey and white matter densities (grey–white ratio; GWR); this quantifies the degree of cerebral oedema. It is reported that a 100% specificity for unfavorable neurological outcomes was achieved between 1 h [60] and up to 72 h [61,62,63] after ROSC.

This bundle of prognostication can be extended by other biomarkers that are currently not recommended such as tau protein, neurofilament light chain, glial fibrillary acidic protein and ubiquitin C-terminal hydrolase-L1, which have recently been described to discriminate between favorable and unfavorable neurological outcomes [64].

In addition to the above-mentioned known and established neurological prognostic options, there is the possibility of whole blood transcriptome analysis [65,66]. One study was able to identify three biomarkers associated with poor neurological outcomes: MAPK3, BCL2 and AKT1 [65]. MAPK3 was significantly up-regulated in patients with poor neurological outcomes. This was further corroborated by multivariate logistic regression, where MAPK3 was the strongest independent predictor for poor neurological outcomes (OR 1.2 [95% CI 1.1 to 16.7]).

Tissier et al. found that HLA family genes were decreased in the group with poor neurological survival [66]. Here, logistic regression was able to predict clinical survival better compared to known person- and setting-specific factors with an area under the curve of 0.94 vs. 0.83, respectively [15]. Organ-specific damage can be assessed using cell-free DNA. This method enables the possibility to assess the potential damage to vital organs such as the brain, heart and lungs. Even if there are no trials in patients with OHCA to date, this method can be used in the future. A limitation can be the availability of expertise to carry out such tests; centralizing these analyses to specialized centers can help overcome this limitation.

These promising first results of additional biomarkers should be studied in future trials, investigating the early prediction of functional neurological outcomes and identifying patients that can benefit from early invasive strategies.

## 3. Discussion

This review assesses the current literature in post-resuscitation care in patients with OHCA. Herein, novel therapy strategies are highlighted and put into the perspective of current guidelines. Different strategies on oxygen and blood pressure levels did not improve survival. Oxygen titration in the initial ROSC phase can be harmful and should be avoided. Beside recent RCTs, observational trials and secondary analyses have shown that hyperoxemia is harmful and associated with an increased mortality rate, especially if PaO_2_ exceeds 300 mm Hg. In patients with reliable pulse oximeter values, the titration of oxygen can be safe if SpO_2_ is >99%, keeping in mind that these patients might deteriorate faster compared to other ventilated patients. Prehospital blood gas analysis can be beneficial in the early phase of ROSC, starting preliminary goal-directed treatment. With the prehospital availability of diagnostic tools, one must not forget that the overarching goal is still to admit the patient to an ICU or emergency department as soon as possible after OHCA. Different aspects of ventilation strategies have been discussed, where a secondary analysis found significantly higher OR for mortality in patients with a higher respiratory rate. Although emerging from a large cohort, these findings should still be interpreted with caution as these data are observational and the original trial focused on TTM [28,42]. Ventilation during OHCA needs further research, especially with respect to different respirators, airway devices and chest compression devices available. These possibilities also include manual chest compression and bag mask ventilation. Whether mild hypercapnia (PaCO_2_ 50–55 mm Hg) within the first 24 h after OHCA is associated with a favorable neurological survival after six months is currently being investigated in the Targeted Therapeutic Mild Hypercapnia after Resuscitated Cardiac Arrest (TAME) trial [67]. These findings can impact future initial ICU treatment and are awaited to be published in June 2023.

The usage of noradrenaline in post-ROSC shock is corroborated, as a matched cohort study showed worse outcomes in patients receiving adrenaline infusion. As this study was not randomized or blinded, whether these results would change in a randomized study must be critically assessed. Further trials might not be impactful. The guidelines currently do not cover the aspect of return of circulation (ROC). As eCPR programs, in-hospital or pre-hospital are currently established around the globe, this challenge must be faced and recommendations for immediate post-ROC treatment are needed. Future studies can address a targeted MAP in the early phase of patients with eCPR.

PCI remains the primary strategy for patients with ST elevation myocardial infarction and in those with hemodynamic instability despite the lack of ST elevations. These recommendations are based on RCTs and have a good certainty of evidence (CoE).

Fever prevention remains critical and is one of the cornerstones in post-resuscitation care, whereas prophylactic antibiotic use does not increase favorable neurological outcomes. Temperature management between 32 and 34 °C was not associated with an increase in favorable neurological outcomes.

Neuroprognostication is made at the patient’s peril, as conformation bias affects our judgement. Therefore, the limitations of current methods should be considered when interpretating biomarkers and rSO_2_. Most of the studies were either retrospective studies, observational trials or secondary analyses, limiting the CoE. Although it may be tempting to start prognosis as early as during the prehospital phase, this underlies certain limitations. RSO_2_ is independent from motion artefacts, making it valid during CPR; however, cerebral blood flow is controlled by PaCO_2_ in patients after OHCA [51].

Novel biomarkers and methods will emerge in the upcoming years and should be considered if feasibility is given outside of clinical studies. Further, a better understanding of their predictive value is needed.

## 4. Conclusions

“*Normo, normo, normo*” is the main goal in post-resuscitation care. Being above the normal ranges has been shown to be harmful in recent studies, especially regarding oxygenation, ventilation and temperature. Strategies to reduce early harm in the treatment of patients with ROSC have yet to be found. Prognostication is still a multi-modal approach, using both technical resources and biomarkers combined.

## Data Availability

Not applicable.

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
