# Peer review of "Developments in Post-Resuscitation Care for Out-of-Hospital Cardiac Arrests in Adults—A Narrative Review"

_jcm, 2023, doi:10.3390/jcm12083009_

Round 1

Reviewer 1 Report

The work is written in an average way. It summarizes the results of research over the past few years. The literature is up to date. 

There is a lack of information on how the literature was selected for review. 

The introduction is very short, I think it would have been worth mentioning the determinants of ROSC in out-of-hospital SCA e.g.:

that the factors associated with ROSC were CPR initiation by witnesses and presence of a shockable rhythm in the scene of accidence. 

https://pubmed.ncbi.nlm.nih.gov/32532201/

https://www.ahajournals.org/doi/10.1161/CIR.0000000000001054 

The discussion is quite short but summarizes the material from the review. 

The narrative review may also have conclusions and I recommend that they be included in the paper. 

Minor editing of English language required

Author Response

Response to Reviewer 1 Comments

Point 1: The work is written in an average way. It summarizes the results of research over the past few years. The literature is up to date.

There is a lack of information on how the literature was selected for review.

The introduction is very short, I think it would have been worth mentioning the determinants of ROSC in out-of-hospital SCA e.g.: that the factors associated with ROSC were CPR initiation by witnesses and presence of a shockable rhythm in the scene of accidence.

https://pubmed.ncbi.nlm.nih.gov/32532201/

https://www.ahajournals.org/doi/10.1161/CIR.0000000000001054

The discussion is quite short but summarizes the material from the review.

The narrative review may also have conclusions and I recommend that they be included in the paper. 

Response 1: We thank the reviewer for their input on our review. We added information on how the literature was selected under the new section 1.1. Further we added the recommended literature to the introduction and lengthened the introduction and discussion. In addition we added a conclusion at the end of the manuscript.

Reviewer 2 Report

I read with great interest he narrative review by Katzenschlager et al. on the most recent literature on post resuscitation care for out of hospital cardiac arrests adult patients.

The review was well conducted and the topic is extremely interesting. However, I have some minor issues to address to the authors:

Line 90. In the EXACT trial that you cited, hypoxic events before ICU admission are indeed reported in Table 3 (OR 2.37 (1.49 to 3.79) p <.001). In fact, the authors state that this could be one of the reasons for the lower survival in the group treated with conservative oxygen. Please modify accordingly.

Line 96. There is more recent evidence on the U-shaped distribution of mortality in OHCA patients. In fact, the recent TTM2 sub-analysis (Robba et al. Crit Care 2022, doi: 10.1186/s13054-022-04186-8) performed a multivariable Cox regression on data from 1418 OHCA patients, thus showing a U-shaped curve of mortality. Moreover, although clinical trials were conducted within the “safe” range of the curve, several observational studies investigated the association between severe hyperoxemia and mortality or neurological outcome, and a recent meta-analysis (La Via et al. Minerva Anestesiologica 2022, doi: 10.23736/S0375-9393.22.16449-7) showed a significant association between a PaO2>300mmHg and a higher mortality and worse neurological outcome in the first 36h after cardiac arrest. Please modify and add these two references.

Lines 107-108. Although this statement is true, the recent secondary analysis of the TTM2 trial (Robba et al. Intensive Care Medicine 2022, doi: 10.1007/s00134-022-06756-4) showed a significant association between respiratory rate and mortality at 6 months. Please modify accordingly.

Line 160. You state “recent trials found no difference in survival” but you only added the TTM2 reference to this sentence. Indeed, previous RCTs have been published on the topic. Please add the results of these two recent meta-analyses of RCTs to validate this statement (Aneman et al. Crit Care 2022, doi: 10.1186/s13054-022-03935-z AND Sanfilippo et al. Journal of Clinical Medicine 2021, doi: 10.3390/jcm10173943).

Author Response

Response to Reviewer 2 Comments

Point 1: I read with great interest he narrative review by Katzenschlager et al. on the most recent literature on post resuscitation care for out of hospital cardiac arrests adult patients.

The review was well conducted and the topic is extremely interesting. However, I have some minor issues to address to the authors:

 Response 1: We thank the reviewer for her/his thoughtful comments and suggested references to increase the value of our paper.

Point 2: Line 90. In the EXACT trial that you cited, hypoxic events before ICU admission are indeed reported in Table 3 (OR 2.37 (1.49 to 3.79) p <.001). In fact, the authors state that this could be one of the reasons for the lower survival in the group treated with conservative oxygen. Please modify accordingly.

Response 2: We thank the reviewer for this comment. In our initial manuscript we reported the OR for hypoxic events of the EXACT trial in line 75 to 77. Line 90 (in the original manuscript) is referring to the BOX Trial by Schmidt et al. We changed the beginning of the sentence in Line 97 to make a clear cut between the two studies. The OR are now stated in lines 93-94.

Point 3: Line 96. There is more recent evidence on the U-shaped distribution of mortality in OHCA patients. In fact, the recent TTM2 sub-analysis (Robba et al. Crit Care 2022, doi: 10.1186/s13054-022-04186-8) performed a multivariable Cox regression on data from 1418 OHCA patients, thus showing a U-shaped curve of mortality.

Response 3: We thank the reviewer for this thoughtful comment. We have implemented the paper by Robba et al. in the lines 111 to 120 accordingly.

Point 4: Moreover, although clinical trials were conducted within the “safe” range of the curve, several observational studies investigated the association between severe hyperoxemia and mortality or neurological outcome, and a recent meta-analysis (La Via et al. Minerva Anestesiologica 2022, doi: 10.23736/S0375-9393.22.16449-7) showed a significant association between a PaO2>300mmHg and a higher mortality and worse neurological outcome in the first 36h after cardiac arrest. Please modify and add these two references.

Response 4: We have changed the beginning of the sentence in line 121 to “Although clinical trials investigating […]” and added the meta-analysis by La Via et al. accordingly.

Point 5: Lines 107-108. Although this statement is true, the recent secondary analysis of the TTM2 trial (Robba et al. Intensive Care Medicine 2022, doi: 10.1007/s00134-022-06756-4) showed a significant association between respiratory rate and mortality at 6 months. Please modify accordingly.

Response 5: We again thank the reviewer for this comment and the literature provided. We have rephrased the sentence in line 140 to “An increase of the respiratory rate during OHCA does not seem to have an effect on PCO2, pH and ROSC rates.” and added the study by Robba et al. accordingly.

Point 6: Line 160. You state “recent trials found no difference in survival” but you only added the TTM2 reference to this sentence. Indeed, previous RCTs have been published on the topic. Please add the results of these two recent meta-analyses of RCTs to validate this statement (Aneman et al. Crit Care 2022, doi: 10.1186/s13054-022-03935-z AND Sanfilippo et al. Journal of Clinical Medicine 2021, doi: 10.3390/jcm10173943).

Response 6: We assume that the reviewer is referring to line 167 in our original manuscript, where TTM is discussed. Line 160 is discussing early PCI in non-STEMI. Therefore, we added the suggested references to the lines 197 to 207.

Round 2

Reviewer 1 Report

Manuscript is ready for publication

no comments